# Proposal of Simplified Standardization of the Cell-Growth-Promoting Activity of Human Adipose Tissue Mesenchymal Stromal Cell Culture Supernatants

**DOI:** 10.3390/ijms25105197

**Published:** 2024-05-10

**Authors:** Shin Enosawa, Sho Kobayashi, Eiji Kobayashi

**Affiliations:** 1Center for Regenerative Medicine, National Center for Child Health and Development, Tokyo 157-8535, Japan; enosawa-s@ncchd.go.jp; 2Kobayashi Regenerative Research Institute, LLC, Wakayama 640-8263, Japan; 3Department of Kidney Regenerative Medicine, The Jikei University School of Medicine, Tokyo 105-8461, Japan

**Keywords:** adipose tissue, mesenchymal stem/stromal cells, conditioned medium, cell growth, cosmetics, efficacy, evidence

## Abstract

The conditioned medium (CM) obtained from mesenchymal stromal cell (MSC) culture has excellent cell growth-promoting activity and is used for cosmetics and healthcare products. Unlike pharmaceuticals, strict efficacy verification is not legally required for these products. However, their efficacy must be substantiated as commercial products. We attempted to simplify CM production and to standardize the evaluation of the growth-promoting activity of CM. CM was obtained through the culturing of two lines of commercially available human adipose tissue-derived MSCs using MEMα with or without 10% fetal bovine serum (FBS) for 24 h. Non-CM control media were produced by the same protocol without MSCs. Growth-promoting activities of the CM were estimated by [^3^H]-thymidine pulse. CM were subjected to molecular weight fractionation with ultrafiltration using 10 k-, 30 k-, 50 k-, and 100 k-membranes. The FBS-free CMs showed 1.34- to 1.85-fold increases and FBS-containing CMs showed 1.45- to 1.67-fold increases in proliferation-promoting activity compared with non-CM controls, regardless of the source of the cell. The thymidine incorporation levels were approximately three times higher in FBS-containing CMs. Aged cells also showed 1.67- to 2.48-fold increases in the activity due to FBS-containing CM, but not to FBS-free CM. The CM activities were sustained even after 1 year at 4 °C. Molecular weight fractionation showed that the activity was recovered in the fraction above 100 k. Clear and stable cell-growth-promoting activity was confirmed with CMs of commercially available adipose tissue MSCs. The activity was detected in the fraction over 100 k. We propose here the importance of standardizing the production and evaluation of CMs to indicate their specific action.

## 1. Introduction

The conditioned medium (CM) obtained from cell cultures has a certain cell growth-promoting potential [1,2,3,4,5]. CM was originally developed as a method for culturing cells that were difficult to culture [6]. To differentiate and proliferate fibroblasts from a single cell, Moen used a very small culture vessel (Carrel’s micro flask) and observed over time without changing the medium. He reported that the cell may have been amplifying some product into its immediate environment which made it favorable for cellular proliferation. Subsequently, CM was found to have not only autocrine but also paracrine effects, such as wound healing, angiogenesis, nerve growth, etc. [2,3,4,5,7,8], and the scope of application is advancing. We have previously studied the use of CM in the preservation solution of pancreatic islets to increase beta cell activities [9]. Along with the recent trend in stem-cell-based regenerative medicine, reports of efficacies of CM in dermatology, such as in the treatment of damaged skin and alopecia, have increased [10,11,12,13,14]. In these studies, autologous, allogeneic, or immortalized MSCs were used. CM is used in cosmetics such as skin lotions [15,16,17,18,19]. Currently, CM is regarded simply as a byproduct of cell culture, and cell therapy regulatory frameworks do not cover its application. Thus, CM is a potentially attractive component for profitable cosmetic and health food items. There is growing interest not only from researchers, but also from clinicians and the biotech industry.

In this communication, we attempt to present the generalized procedure of CM production and estimation for the following purposes: (1) to demonstrate a precise, simple method specified for cell density, medium type and volume, incubation time, etc., and the cytochemical characteristics of CM to people who have become interested in CM; (2) to elucidate the important factors to assure reproducibility, including the setting of the control medium; and (3) to propose the importance of preclinical validation. The first and second objectives are interrelated; a literature search review pointed out that the cell culture conditions such as passage number, seeding density, conditioned media harvesting time, functional uptake, and working concentration were poorly documented [19]. Although those items are basic and typical and skilled researchers may not be interested, they are crucial to secure the reproducibility of the experiment. The third issue involves the licensing system. While cell-processed products are approved under the laws in Japan [20], CM products that belong to cosmetic and health food items can be commercialized without prior experimental testing and post-marketing verification. We recently disputed the importance of efficacy testing in cosmetic and health food items [21]. For these purposes, we propose a simple standardized method of the production and estimation of CM.

## 2. Results

### 2.1. Construction of Protocols for CM Production and Cell-Proliferation Assay

We constructed a simple protocol to establish the production and estimation of CMs from human adipose tissue MSCs (Figure 1a,b; for experimental details, see Section 4.1). In CM production, the total thawed cells recovered from a cryopreservation tube (approximately 1 × 10^6^ cells) were directly seeded on a 150 mm-diameter dish and cultured for 96 h with the FBS-containing MEMα medium to ensure the adhesion and restore the proliferative potential of the cells that were impaired by cryopreservation (Figure 1a). To produce CMs, the dish was thoroughly washed with saline and the medium changed for CM with FBS-free or FBS-containing MEMα. A 150 mm diameter dish (practical cultivation area: 151 cm^2^) was used to obtain the necessary volume of CM (25 mL) to conduct the experiment and to avoid cell passaging, which could cause experimental errors. To eliminate the influences of the degradation of bioactive substances during incubation and the possibility of residual FBS after pre-culture, the same operation was performed in a control dish without cells to obtain a non-CM control medium. The phase contrast images show the increase in KW-4209 cells from 2 h after seeding to the time of collection of FBS-containing CM. Since the area of this field of view is 0.27 mm^2^, the photo from 2 h after seeding is almost consistent with the calculated cell density (Table 1). The cells at the time of CM collection were proliferated with intact morphology. The glucose concentration was slightly decreased in non-CM controls and FBS-free (85.5 ± 0.5 and 82.0 ± 2.0 mg/dL, mean ± range, *n* = 2, respectively). The concentration decreased most to 70.5 ± 1.5 mg/dL, suggesting the effect of cell proliferation. Since the decrease was not likely to affect the cell culture, however, the following cell proliferation assay was performed without balancing glucose concentration.

The cell proliferation was assayed by [^3^H]-thymidine incorporation (Figure 1b). Thawed cells were counted for viability and cultured with FBS-containing MEMα in the first 2 h to ensure adhesion. Then, the medium was changed to the test CMs and [^3^H]-thymidine was added. Because of the low cell number, the pulse duration was set slightly longer (46 h) than usual.

The cellular indices before and after CM production are summarized in Table 1. The initial cell viability and viable cell number at cell thawing were 88.0% and 1.26 × 10^6^ and 88.9% and 0.99 × 10^6^ in #PT-5006 and KW-4209 cells, respectively. To produce CMs, the cells were seeded at a density of 7318 cells/cm^2^ and 6310 cells/cm^2^ on a 150 mm diameter dish with FBS-containing MEMα medium. The cell densities at the end of CM production were increased as follows: 12,583 and 15,232 cells/cm^2^ with FBS-free and FBS-containing CM of #PT-5006 cells, and 20,861 and 27,881 cells/cm^2^ with FBS-free and FBS-containing CM of KW-4209 cells. As indicated by fold increases, the cells cultured with FBS-containing medium throughout the culture increased 1.24 or 1.27 times higher than those cultured with FBS-free medium in the last 24 h. Cell viabilities were around 95% under all conditions.

### 2.2. Cell-Proliferation-Promoting Activities of CMs

All CMs showed cell proliferation-promoting activities regardless of the combination of test cells and CM source (Figure 2). FBS-free and FBS-containing CMs obtained from #PT-5006 and KW-4209 cells showed 1.48- and 1.67-fold, and 1.45- and 1.67-fold higher activities than the non-CM control medium, respectively, although data with FBS-free CM were not statistically significant except those for the #PT-5006 cells and FBS-containing KW-4209 CM combination (Figure 2). In terms of [^3^H]-thymidine uptake levels, cells treated with FBS-containing CM showed the highest uptake, followed by FBS-containing non-CM control medium, serum-free CM, and serum-free non-CM control medium. The activities of CM were detectable for the other cells compared to those from which the CM was produced. The responsiveness of cell proliferation was better with #PT-5006 throughout the experiment.

### 2.3. Cell-Proliferation-Promoting Activities of CM after 1 Year of Refrigerated Storage

The proliferation-promoting activities of CMs were detectable after 1 year of refrigerated storage (Figure 3). Unfortunately, the differences were not statistically significant with KW-4209 cells, of which the responsiveness of cell proliferation was low, as described above (Figure 2).

### 2.4. Cell-Proliferation-Promoting Activities of CM with Young and Aged Cells

The thymidine incorporation of young and aged KW-4209 cells produced results of 3865 ± 1272 cpm and 371 ± 332 cpm (mean ± SD, *n* = 10) with non-CM FBS-free and non-CM FBS-containing control medium, respectively, suggesting that the aged cells showed low proliferative activities (10.1% and 29.5%) (Figure 4). Aged cells were cultured for 14 days with two passages, and the cell division number was 7.25 ± 1.15 (mean ± SD, *n* = 4, experimental details see also Section 4.3). FBS-free CM had no growth promoting effect on aged cells (0.85 ± 0.29-fold), whereas FBS-containing CM had a 2.30 ± 0.50-fold promoting effect.

### 2.5. Cell-Proliferation-Promoting Activities of CM Fractionated by Molecular Weight

CM fractionated by molecular weight using limiting membranes showed little activity in the fraction with a molecular weight under 100,000 in both young and aged cells, suggesting that the activity was in the fraction over 100 k (Figure 5).

## 3. Discussion

Even after the development of various synthetic media, cell culture supernatants have been used for cells that are difficult to culture. Therefore, the culture supernatant or conditioned medium (CM) has been a large topic in cell biology. The nature of CM activity is still not well understood, but in recent years both bioactive proteinaceous factors and exosomes have been attracting attention, and the term secretome has been coined [1].

CM has been used for cosmetic purposes worldwide without an in-depth study of its biological effects. Similar efficacy has been reported with platelet-rich plasma or platelet lysate [22], but blood derivatives are inseparable from the risk of infection. In this respect, CM that can be produced in a controlled process yields a constant quality product. Recently, CM was shown to suppress the levels of type 2 cytokines and chemokines, including thymic and activation-regulated chemokines (TARCs), TNF-α, and IL-6 levels in HaCaT cells [23]. Further, CM also suppressed the levels of IL-4 and IL-13 in Th2 cells, suggesting that CM may have therapeutic potential as a drug for atopic dermatitis. It is necessary to determine whether the results obtained with laboratory-scale culture are identical to those obtained with mass culture for commercial use. CM from human umbilical cord-derived MSC culture obtained using laboratory flasks and a mass culture bioreactor have been compared for effects on ultraviolet B-induced oxidative stress as well as anti-aging and melanogenic properties [24]. Quantitative evaluation has also been performed for growth factors, hyaluronic acid, procollagen, promelanin, and cell viability. They also measured the induction of hyaluronic acid and collagen synthesis in CCD-986SK cells and the melanogenic activity in SK-MEL-31 cells based on melanin content and tyrosinase activity. The authors concluded that their bioreactor-derived CM showed active ingredients; however, as their measurement was complex and non-standardized, a comparison with other CMs was difficult.

Although the analysis of CM has been conducted diligently, the question of whether the nature of CM action is due to bioactive molecules or exosomes is still controversial. A recent report indicated that both had specific biological efficacy to facilitate tissue repair using an in vivo adipogenesis test [25]. Studies focusing on exosomes have shown that they promote neuronal differentiation and enhance neutrophil function [26,27]. On the other hand, a bioactive protein was identified in the CM from adipose MSC [8,28] and some reports have shown molecular mechanisms of CM effects [29,30]. Interestingly, the secretome profile was different for each tissue from which MSCs were obtained, although no significant differences in cell biological effects were observed [31]. Along with the advances in analytical techniques, reports of comprehensive analyses are increasing. From this point of view, it is important to provide detailed descriptions of the conditions of the preparation and activity assay of CMs, as well as the standardization of the methods.

Our proposal of a simple in vitro protocol for the production and assay of CMs enables a fair comparison of the effect of CMs. The present protocol was established with particular attention paid to the simplicity and availability of the control medium: (1) both CM preparation and the growth assay can be started directly from cryopreserved cell lines without passaging; (2) relatively large amounts (25 mL) of CM can be obtained in a single culture; (3) exactly the same procedure can be set to obtain the control medium to eliminate the effects of degradation of bioactive substances during culture and the possibility of FBS remaining in the initial control medium. In this study, we examined the CM’s autocrine and paracrine effects on the cells in the same category as a representative indicator. Although the effects on other cells, such as epidermal and dermal cells, should also be examined, it would be practical to screen effective CMs using MSCs as test cells. In addition, it is also necessary to confirm their characteristics as MSCs, e.g., in the maintenance of surface antigens and differentiation activity. Further detailed studies are expected in the future.

We found that both FBS-free and -containing CMs showed growth-promoting activity across cell types. Notably, FBS-containing CM was still active when refrigerated for 1 year. However, the activity of FBS-free CM disappeared. This could be attributed to the stabilizing effect of proteins or lipids by FBS. It will be necessary to develop protective agents to replace serum in future studies. The activity was recovered in a high molecular weight fraction. This may support the fact that the CM factor produces large proteinaceous molecules or exosomes. Unfortunately, the latter half of this study could not be performed with #PT-5006, which is highly reactive to cell-proliferative activities of CMs rather than KW-4209, due to unavailability. This time, we used radioactive thymidine as a tracer for the cell proliferation assay, but other methods such as a colorimetric MTT assay could be substituted, while slightly more cells are necessary for the assay. In any case, the method presented here is useful for selecting superior CM lots. This standardized simple assay system could facilitate the development of CM applications.

## 4. Materials and Methods

### 4.1. Production of CMs

Frozen human adipose-derived stromal cells (#PT-5006, Poietics™, Lonza, Basel, Switzerland) and human pre-adipocytes (KW-4209, LIFELINE, KURABO, Osaka, Japan) were purchased. The experimental use of these commercially available human cells is not subject to the Japanese Ethical Guidelines for Medical and Health Research Involving Human Subjects. The culture medium was MEMα (135-15175, Fuji Film Wako, Tokyo, Japan) and antibiotics, penicillin (1000 IU/mL)-streptomycin (100 μg/mL)-amphotericin B (250 ng/mL) (161-23181, Fuji Film Wako, Tokyo, Japan) and kanamycin sulfate (100 μg/mL) (117-00961, Fuji Film Wako, Tokyo, Japan) were added to the medium throughout the experiment. Frozen cells (1 × 10^6^ cells) were thawed and seeded on a 150 mm diameter dish (430599, Corning cell culture dish, Corning, NY, USA) with 25 mL of 10% FBS (SH30070.03, HyClone, Cytiva, Marlborough, MA, USA)-containing MEMα (Figure 1a). After 96 h culture at 37 °C under 5% CO_2_, cells were washed twice with 30 mL of saline and the medium was changed to 25 mL of serum-free or FBS-containing MEMα and cultured for 24 h to obtain CMs. The non-conditioned control medium was prepared by the same procedure without cells. The resulting CM and control media did not become turbid or form sediments even after 1 year of refrigerated storage and were used directly for culture without filtration or centrifugation.

### 4.2. Assay of Cell-Proliferation-Promoting Activities

To estimate the growth-promoting activity of CM, #PT-5006 or KW-4209 cells were thawed and cultured with 10% FBS-containing MEMα for 2 h at 1.5 × 10^4^ live cells/0.1 mL/well, 4.7 × 10^4^ live cells/cm^2^ (4 wells each; 3599, Corning 96-well cell culture plate; practical cultivation area, 0.32 cm^2^) at 37 °C under 5% CO_2_. Then, cells were washed twice with 0.5 mL of saline and the medium was changed to FBS-free or FBS-containing test CMs and 37 kBq of [^3^H]-thymidine was added (NET027Z001MC, PerkinElmer, Waltham, MA, USA) (Figure 1b). Forty-six hours after the medium change, the cells were harvested with trypsin–EDTA solution (203-20251, Fuji Film Wako, Tokyo, Japan) and the incorporated radioactive thymidine was quantified using a liquid scintillation spectrometer (Top-Count NXT, PerkinElmer). Cell viability was determined by the trypan blue exclusion test (Gibco™ Trypan Blue Solution, Thermo Fisher Scientific, Waltham, MA, USA).

### 4.3. Preparation of Aged Cells, CM Fractionation, and Glucose Determination

Aged cells were first cultured for 5 days, and after harvest, cells were diluted 2-fold and cultured for another 9 days with 25 mL of 10% FBS-containing MEMα in 150 mm-diameter dishes. After passage, the medium was changed on culture day 4. The cells were harvested using trypsin-EDTA solution and stored with cryopreservation solution (STEM CELL BANKER, ZENOGEN PHARMA CO., LTD., Fukushima, Japan) in liquid nitrogen until use.

The CM was fractionated using the Amicon Ultra-2 mL (10 k-, 30 k-, 50 k-, and 100 k-device, Merck-Millipore, Burlington, MA, USA). Cell-proliferation-promoting activities were determined from three to eight independent plates.

Glucose concentration was determined colorimetrically with a clinical test kit (LabAssay™ Glucose, 638-50971, Fuji Film Wako, Tokyo, Japan).

### 4.4. Statistics

Statistical significance was tested using ANOVA followed by Fisher’s protected least significant difference (PLSD) test (JMP11, SAS Institute, Cary, NC, USA, https://www.sas.com/en_us/home.html, accessed on 2 April 2024); *p* < 0.05 was considered statistically significant.

## 5. Conclusions

We developed a simplified protocol of CM production and assay for its cell-proliferation promoting activity. According to our method, the activity of CM was detectable regardless of the cell source, when maintained in refrigerated storage for 1 year, and when kept in a fraction of over 100 k. Standardizing the protocol will expand the base of CM research and ensure the reliability of non-medical CM products such as cosmetics that are not subject to rigid regulations worldwide.

## Figures and Tables

**Figure 1 ijms-25-05197-f001:**
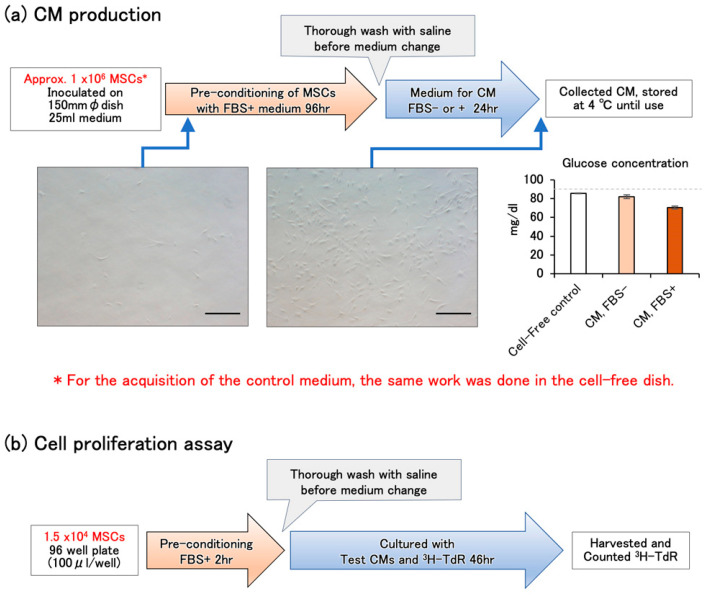
(**a**) Schematic illustration of CM production. The total thawed cells recovered from a cryopreservation tube (approximately 1 × 10^6^ cells) were directly seeded on a 150 mm-diameter dish at a density as shown in Table 1 (#PT-5006: 7318 ± 99 cells/cm^2^, KW-4209: 6310 ± 310 cells/cm^2^) and cultured for 96 h with the FBS-containing MEMα medium. Then, the medium was changed to produce CM. The photos show phase contrast images of cultured KW-4209 cells at the points indicated by the arrows 2 h after seeding and the time of collection of FBS+ CM. The bars indicate 100 μm (original magnification; 200×). The glucose concentration in the medium is shown as the mean ± range of the two lots, respectively. The gray dashed line indicates the initial glucose concentration of the MEMα medium. (**b**) Schematic illustration of CM production. Thawed cells were cultured with FBS-containing MEMα for 2 h to ensure the adhesion, the medium was changed to test CMs, and [^3^H]-thymidine was added.

**Figure 2 ijms-25-05197-f002:**
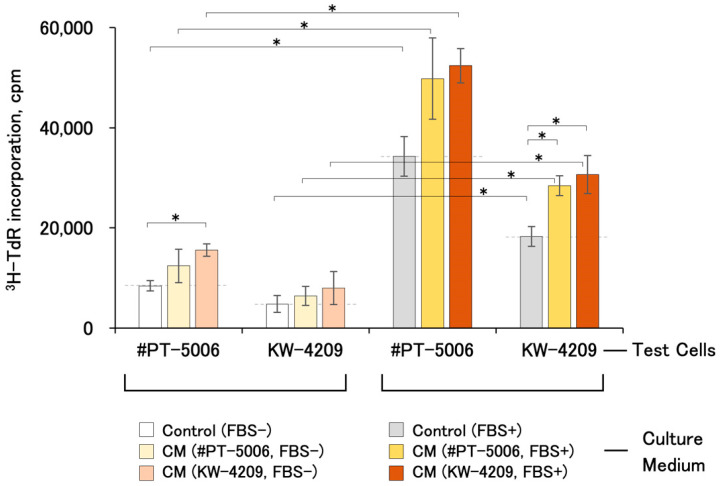
Cell proliferation-promoting activities of various CMs. Each CM and respective non-CM control medium were obtained by the method shown in Section 4 and Figure 1. Briefly, data in the left and right columns were from FBS-free and FBS-containing CMs, respectively. A cell proliferation assay was performed according to the procedure written in Section 4 and Figure 1b. Asterisks indicate statistical significance, with *p* < 0.05 from ANOVA and Fisher’s protected least significant difference (PLSD) test.

**Figure 3 ijms-25-05197-f003:**
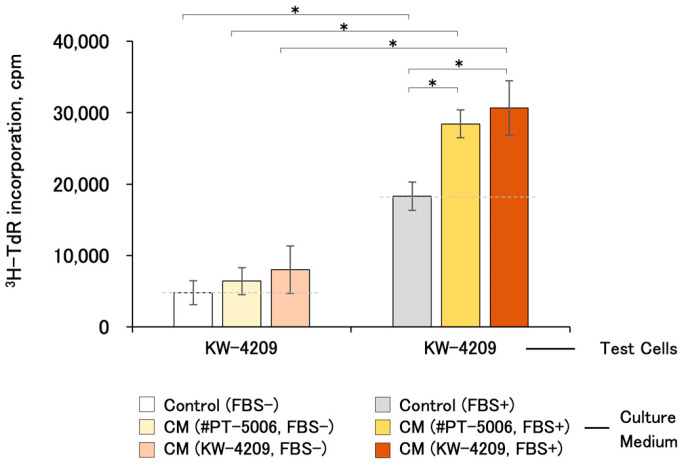
Cell proliferation-promoting activities of CM after 1 year of refrigerated storage. Due to the unavailability of cells, the experiment was performed only with KW-4209. Gray dashed line indicates the activity of non-CM control medium. Asterisks indicate statistical significance; *p* < 0.05 by ANOVA and Fisher’s protected least significant difference (PLSD) test.

**Figure 4 ijms-25-05197-f004:**
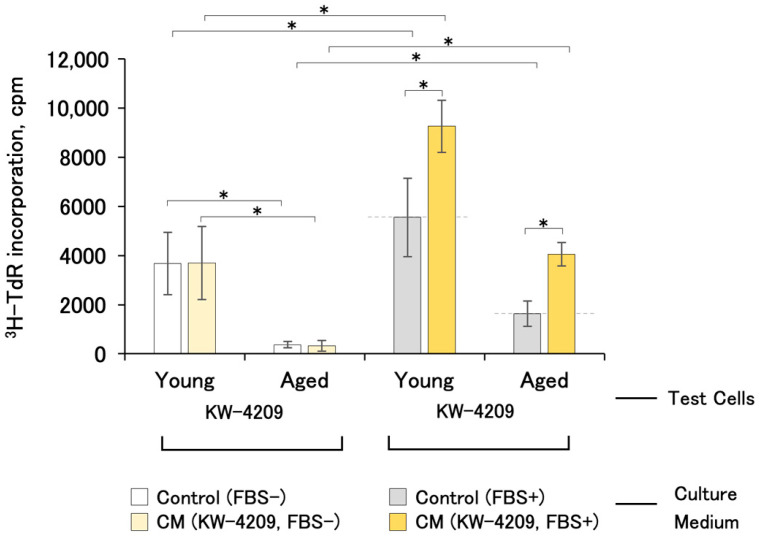
Cell-proliferation-promoting activities of CM with young and aged cells (*n* = 4 and 8, respectively). Due to the unavailability of cells, the experiment was performed only with KW-4209. Gray dashed line indicates the activity of non-CM control medium. Asterisks indicate statistical significance; *p* < 0.05 according to ANOVA and Fisher’s protected least significant difference (PLSD) test.

**Figure 5 ijms-25-05197-f005:**
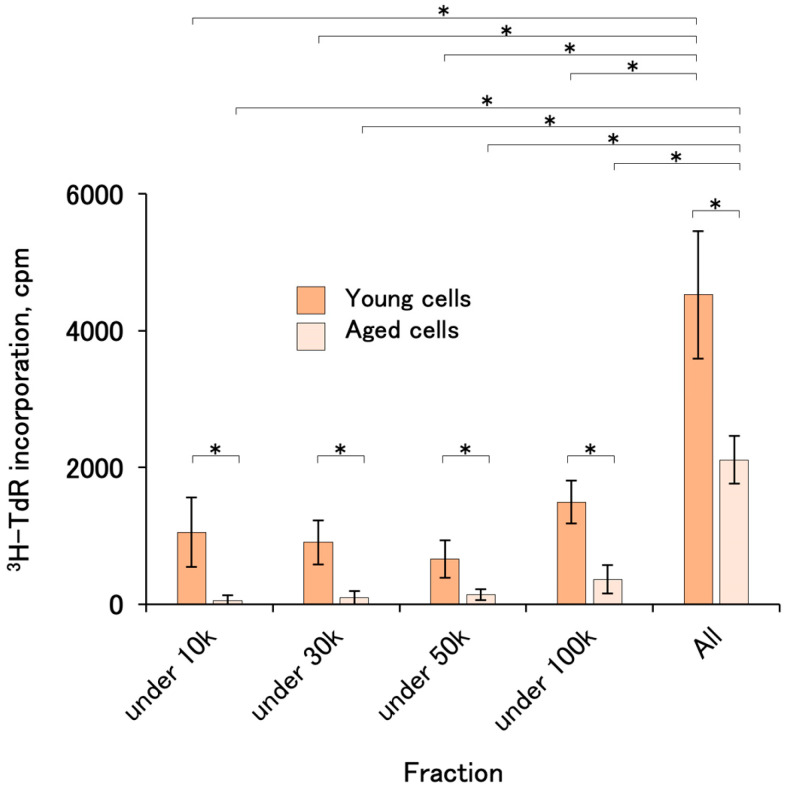
Cell-proliferation-promoting activities of CM fractionated by molecular weight. FBS-containing CMs from #PT-5006 and KW-4209 were mixed and fractionated with limiting membranes. The activities were assayed with KW-4209 cells. In the preliminary experiment, the activities were not recovered with FBS-free CM. Asterisks indicates statistical significance; *p* < 0.05 according to ANOVA and Fisher’s protected least significant difference (PLSD) test.

**Table 1 ijms-25-05197-t001:** Characteristics of used cells and changes in cellular indices.

	#PT-5006, Human adipose derived stem cells ^(1)^	KW-4209, Human pre-adipocytes ^(1)^
Source ^(1)^	Human adipose tissue obtained as lipoaspirate from patients undergoing elective liposuction surgery procedures.	Adult adipose tissue
Passage number at shipment ^(1)^	1	2
Cell viability, % ^(2)^	88.0 ± 9.4, *n* = 5	88.9 ± 4.8, *n* = 12
Viable cell number (×10^6^) per tube ^(2)^	1.26 ± 0.24, *n* = 5	0.99 ± 0.28, *n* = 12

Cellular changes in CM production process
Initial cell density (/cm^2^)	7318 ± 99, *n* = 2 ^(3)^	6310 ± 310, *n* = 4
Medium for CM ^(4)^	FBS-	FBS+	FBS-, *n* = 2	FBS+, *n* = 2
Cell viability, %	96.2	97.9	95.1 ± 3.8	93.1 ± 6.1
Cell density (/cm^2^) at the time of harvest	12,583	15,232	20,861 ± 728	27,881 ± 1788
Fold increase ^(5)^	1.70	2.11	3.40 ± 0.26	4.32 ± 0.38

^(1)^ Cell nomenclature, source, and passage number were from the manufacturers’ brochure. Recommended seeding density is around 5000 cells/cm^2^ (can be changed depending on the experiment), and quality assurance is up to 4 passages. ^(2)^ All data of cell thawing in the experiment. Mean ± SD. ^(3)^ In the case of *n* = 2, data are expressed as mean ± range. ^(4)^ The medium used for the production of CM is indicated. FBS-, FBS-free MEMα medium; FBS+, FBS-containing MEMα medium. ^(5)^ The number obtained by cell density at harvest divided by initial cell density.

## Data Availability

Data contained within the article are included in the text, and further inquiries can be directed to the first or corresponding author.

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
