# Peer review of "Proposal of Simplified Standardization of the Cell-Growth-Promoting Activity of Human Adipose Tissue Mesenchymal Stromal Cell Culture Supernatants"

_ijms, 2024, doi:10.3390/ijms25105197_

Round 1
Reviewer 1 Report
Comments and Suggestions for Authors
The article authored by Shin Enosawa and colleagues, titled "Proposal of Standardization of the Cell Growth-Promoting Activity of Human Adipose Tissue Mesenchymal Stromal Cell Culture Supernatants," aims to establish guidelines or a protocol for the production of conditioned media (CM) from adipose tissue stem cells.
The number of starting cells is crucial for understanding cell viability. On line 64, I'm unsure because upon reviewing the materials and methods section, it appears that the authors began with one million cells. Equally important is knowing the recovery rate and the seeding density (n cells/cm2) at which the cells are plated.
At what passage were the cells used? What was their senescence rate?
There is a lack of information regarding adolescent and senescent cells. How were they obtained? Even in the case of obtaining senescent cells, how were they defined? Positive controls obtained through irradiation or exposure to H2O2, SAHA, or 5Aza would be essential. Subsequent evaluation of p21, p16, and beta galactosidase assay would be minimal requirements to characterize senescent cells.
That being said, numerous data are missing that would allow for the standardization of CM production methods. Therefore, I believe the manuscript is not acceptable in its current form.
Comments on the Quality of English Language
No comment
Author Response
To Reviewer 1
The article authored by Shin Enosawa and colleagues, titled "Proposal of Standardization of the Cell Growth-Promoting Activity of Human Adipose Tissue Mesenchymal Stromal Cell Culture Supernatants," aims to establish guidelines or a protocol for the production of conditioned media (CM) from adipose tissue stem cells.
[Answer] Thank you very much for your invaluable comments. We did our best to revise the manuscript. The reviewer’s comments revealed that we did not mention our proposal well. We have clarified this point throughout the manuscript. We emphasized the simplicity and standardization of this method in the Title and the text [Line 2, 55, 68, 231-234].
The number of starting cells is crucial for understanding cell viability. On line 64, I'm unsure because upon reviewing the materials and methods section, it appears that the authors began with one million cells. Equally important is knowing the recovery rate and the seeding density (n cells/cm2) at which the cells are plated.
[Answer] I am very sorry not to mention the important items. We added Table 1 to demonstrate important items for cell culture including cell density and added description in the text [Line 130-137, 119-129]
At what passage were the cells used? What was their senescence rate?
[Answer] Thank you very much for your important comment. We mentioned the passage numbers in Table 1 (#PT-5006, passage 1; KW-4209, passage 2 at the time of shipment). [Line 130-137]
There is a lack of information regarding adolescent and senescent cells. How were they obtained? Even in the case of obtaining senescent cells, how were they defined? Positive controls obtained through irradiation or exposure to H2O2, SAHA, or 5Aza would be essential. Subsequent evaluation of p21, p16, and beta galactosidase assay would be minimal requirements to characterize senescent cells.
[Answer] I am sorry but we have not taken data on surface antigens or staining to determine senescence. Thus, we changed the word, ‘senescence’ to ‘aged’, and accordingly, changed the word, ‘adolescent’ to ‘young’ throughout the manuscript including Figures 4 and 5. The data we have are passage number, division number, and decrease of thymidine uptake, i.e., 1) aged cells underwent one passage and 7.25 ± 1.15 of cell division after thawing, 2) thymidine uptake was decreased from 3,865 ± 1,272 (young cells) to 371 ± 332. We added the description about these data in the text. [Line 168-173]
That being said, numerous data are missing that would allow for the standardization of CM production methods. Therefore, I believe the manuscript is not acceptable in its current form.
[Answer] We apologize for the inadequate description of the first edition. We did our best to revise the manuscript according to your careful comments. We would be grateful if you reconsider whether the manuscript is worth publishing in IJMS as Communication.
In addition, the manuscript has been checked by MDPI's English editing service.

Reviewer 2 Report
Comments and Suggestions for Authors
Dear Authors,
I have recently reviewed the manuscript entitled "Proposal of standardization of the cell growth-promoting activity of human adipose tissue mesenchymal stromal cell culture supernatants" and below you will find my comments for this manuscript. The manuscript indeed may be attractive for the scientists in the field, however revisions are required before the manuscript further processed.
1) In the introduction section, the main scope of the manuscript is not clearly presented. The authors indicated the use of the conditioned medium derived from MSCs for cosmetic purposes, however, which cosmetic purposes this medium can be truly used. The authors need further clarification regarding the main aim of the manuscript.
2) In materials and methods, the authors indicated the use of human adipose derived stromal cells and pre-adipocytes. However it is not very clear for the readers which of these cells were used for the production of CM.
3) Furthermore, the author should indicate in the materials and methods section the number of different cell batches used for the production of the CM.
4) From the results presented in figure 2, CM+FBS have supperior capacity for inducing the cell proliferation of adipose derived stem cells and pre-adipocytes compared to the CM-FBS. However, comparison of the cell proliferation to obtain the statitistically significant differences may be performed also between CM+FBS and CM-FBS cultured cells.
5) The authors should also perform a proteomic approach to characterize the CM-FBS medium, to clearly know the growth factors, hormones and proteins that may be responsible for the cell expansion.
6) In general, i think if the CM is intended to be used for cosmetic purposes also, other cell types should be tested, e.g. keratinocytes, epithelial and endothelial cells.
6) Do the authors tested besides the cell proliferation, other parameters regarding the stem cell properties, such as the differentiation capacity or the immunophenotype of the cells?
7) Due to the reason that in cell culture, the scientists are making effort to use animal derived free media, what was the purpose of using CM+FBS.
8) Is the MSCs derived CM superior to platelet rich plasma or platelet lysate. The authors should discuss this.
9) Minor corrections, There are two Figures 2 in the results section. Also, microscopic images of cultured cells with CM+FBS, CM-FBS and control should also be provided.
Author Response
To Reviewer 2
I have recently reviewed the manuscript entitled "Proposal of standardization of the cell growth-promoting activity of human adipose tissue mesenchymal stromal cell culture supernatants" and below you will find my comments for this manuscript. The manuscript indeed may be attractive for the scientists in the field, however revisions are required before the manuscript further processed.
Thank you for your careful reading and invaluable comments. We have done our best to address them as follows.
1) In the introduction section, the main scope of the manuscript is not clearly presented. The authors indicated the use of the conditioned medium derived from MSCs for cosmetic purposes, however, which cosmetic purposes this medium can be truly used. The authors need further clarification regarding the main aim of the manuscript.
[Answer] We are very sorry for the lack of references. We added relevant references for cosmetic use of CMs. [Line 46 and references 15-19]
2) In materials and methods, the authors indicated the use of human adipose derived stromal cells and pre-adipocytes. However, it is not very clear for the readers which of these cells were used for the production of CM.
[Answer] This time we used commercially available primary cell lots, #PT-5006 and KW-4209. To clarify the experimental details, we added Table 1. [Line 130-137]
3) Furthermore, the author should indicate in the materials and methods section the number of different cell batches used for the production of the CM.
[Answer] We are very sorry for the insufficient description. For CM production, both #PT-5006 and KW-4209 were used immediately after thawing. One of our aims was to establish a simple protocol, thus we did not passage in the preparation of CMs. This point was more clarified with the addition of Table 1 and description in the Discussion. [Line 130-137, 233]
4) From the results presented in figure 2, CM+FBS have superior capacity for inducing the cell proliferation of adipose derived stem cells and pre-adipocytes compared to the CM-FBS. However, comparison of the cell proliferation to obtain the statistically significant differences may be performed also between CM+FBS and CM-FBS cultured cells.
[Answer] As you pointed out, the FBS-containing medium was more active than the serum-free medium in all groups. We added the indication of statistical significance in Figure 2 and legend. We also added the indication of statistical significance in Figures 3, 4, and 5. [Line 150-156, 162-166, 176-180, 186-190]
5) The authors should also perform a proteomic approach to characterize the CM-FBS medium, to clearly know the growth factors, hormones, and proteins that may be responsible for the cell expansion.
[Answer] As you pointed out, we should have analyzed the nature of cell proliferation activity but did not achieve it. This time, we focused our aim on establishing an experimental format. We have added a sentence to the Discussion to clarify this point. [Line 231-237]
6) In general, I think if the CM is intended to be used for cosmetic purposes also, other cell types should be tested, e.g. keratinocytes, epithelial and endothelial cells.
[Answer] As you pointed out, we should have examined the CM effects on the constituent cells of the skin but did not achieve them. In this study, we examined the effects of autocrine and paracrine on the cells and cells in the same category as a representative indicator. We have added this point to the text. We hope that detailed studies will be developed from this experimental scheme. [Line 237-240]
7) Do the authors tested besides the cell proliferation, other parameters regarding the stem cell properties, such as the differentiation capacity or the immunophenotype of the cells?
[Answer] As you pointed out, the stemness and differentiation capability of proliferated cells by CM are important issues but we did not determine them. We have added a note in the text that we have only examined proliferation in this study, but we hope that more detailed studies will be developed from this experimental platform in the future to examine how CM affects stemness and differentiation directionality. [Line 240-242]
8) Due to the reason that in cell culture, the scientists are making effort to use animal derived free media, what was the purpose of using CM+FBS.
[Answer] As you pointed out, we had aimed to obtain the CM effects with FBS-free medium and we used FBS-containing medium as a positive control. However, the activity of FBS-free CM was less marked than FBS-containing CM. In addition, refrigerated shelf life was also poor. We discussed the protective effect of serum and the need of the development of some protective agents to replace serum in future studies. [Line 245-247]
9) Is the MSCs derived CM superior to platelet rich plasma or platelet lysate. The authors should discuss this.
[Answer] Platelet rich plasma or platelet lysate derived from human sources is an infectious disease risk. CM allows for production in a quality-controlled process. We added this to the discussion. [Line 199-201]
10) Minor corrections, there are two Figures 2 in the results section. Also, microscopic images of cultured cells with CM+FBS, CM-FBS and control should also be provided.
[Answer] Thank you for careful reading. We have corrected the figure number. Since we focused on thymidine uptake, only the microscope images of KW-4209 cells were available. We added them in Figure 1. [Line 104-115]
PS. the manuscript has been checked by MDPI's English editing service.

Round 2
Reviewer 2 Report
Comments and Suggestions for Authors
Dear Authors,
You have succesfully addressed the majority of my comments. Well done!!